# Derivation and Characterization of Immortalized Human Muscle Satellite Cell Clones from Muscular Dystrophy Patients and Healthy Individuals

**DOI:** 10.3390/cells9081780

**Published:** 2020-07-26

**Authors:** Jimmy Massenet, Cyril Gitiaux, Mélanie Magnan, Sylvain Cuvellier, Arnaud Hubas, Patrick Nusbaum, F Jeffrey Dilworth, Isabelle Desguerre, Bénédicte Chazaud

**Affiliations:** 1Institut NeuroMyoGène, Université Claude Bernard Lyon 1, Université de Lyon, CNRS 5310, INSERM U1217, 69008 Lyon, France; jimmy.massenet@univ-lyon1.fr; 2Institut Cochin, INSERM U1016, CNRS UMR8104, Université Paris Descartes, Université Sorbonne Paris Cité, 75014 Paris, France; cyril.gitiaux@aphp.fr (C.G.); melanie.magnan@inserm.fr (M.M.); sylvain.cuvellier@ibgc.cnrs.fr (S.C.); isabelle.desguerre@aphp.fr (I.D.); 3Laboratoire de Culture Cellulaire, Service de Génétique et Biologie Moléculaires—Hôpital Cochin, APHP.5, Assistance Publique-Hôpitaux de Paris, 75014 Paris, France; arnaud.hubas@aphp.fr (A.H.); patrick.nusbaum@aphp.fr (P.N.); 4Sprott Center for Stem Cell Research, Ottawa Hospital Research Institute; Department of Cellular and Molecular Medicine, University of Ottawa, Ottawa, ON K1H 8M5, Canada; jdilworth@ohri.ca

**Keywords:** human muscle stem cells, immortalization, degenerative myopathies, Duchenne muscular dystrophy, congenital myopathies

## Abstract

In Duchenne muscular dystrophy (DMD) patients, absence of dystrophin causes muscle wasting by impacting both the myofiber integrity and the properties of muscle stem cells (MuSCs). Investigation of DMD encompasses the use of MuSCs issued from human skeletal muscle. However, DMD-derived MuSC usage is restricted by the limited number of divisions that human MuSCs can undertake in vitro before losing their myogenic characteristics and by the scarcity of human material available from DMD muscle. To overcome these limitations, immortalization of MuSCs appears as a strategy. Here, we used CDK4/hTERT expression in primary MuSCs and we derived MuSC clones from a series of clinically and genetically characterized patients, including eight DMD patients with various mutations, four congenital muscular dystrophies and three age-matched control muscles. Immortalized cultures were sorted into single cells and expanded as clones into homogeneous populations. Myogenic characteristics and differentiation potential were tested for each clone. Finally, we screened various promoters to identify the preferred gene regulatory unit that should be used to ensure stable expression in the human MuSC clones. The 38 clonal immortalized myogenic cell clones provide a large collection of controls and DMD clones with various genetic defects and are available to the academic community.

## 1. Introduction

Muscular dystrophies are a group of inherited incurable myopathies characterized by ongoing rounds of skeletal muscle degeneration and regeneration. Duchenne muscular dystrophy (DMD) is caused by mutations in the dystrophin gene, which lies on the X chromosome, where the loss of a functional dystrophin protein causes muscle wasting with an incidence of 1 in 5000 to 1/6000 boys [1]. In these patients, dystrophin protein deficiency causes recurring myofiber lesions, leading to the continuous activation of the regeneration process that is sustained by muscle stem cells (MuSCs). The ongoing regeneration eventually leads to the exhaustion of MuSC capacities to repair the muscle. Over time, progressive muscle wasting and increased fibrosis lead to skeletal muscle loss of function [2]. In order to understand the cellular and molecular mechanisms involved in DMD, mouse models are commonly used. The mdx mouse model, which bears a mutation in the dystrophin gene, presents a mild clinical phenotype when compared to the human pathology due to several factors, including compensation by the related subsarcolemmal protein utrophin and the presence longer telomeres in murine MuSCs [3,4]. Past studies have used human MuSCs isolated from skeletal muscle of DMD patients to understand the mechanisms leading to MuSC exhaustion. Indeed, MuSCs isolated from healthy muscle biopsies recapitulate adult myogenesis in vitro, including activation, proliferation, differentiation and eventual myoblast fusion [5]. Initial attempts to characterize the behavior of DMD-derived MuSCs encountered two major issues. The first is that MuSC cultures were not pure, as they also contained non-myogenic cells. This issue has been overcome by the use of cell sorting techniques to obtain pure populations of myogenic cells. For this purpose, primary human MuSCs are usually sorted based on their expression of CD56 (NCAM) on their cell surface. In human skeletal muscle, CD56 is expressed by quiescent satellite cells, activated myoblasts and young regenerating myofibers [6]. The second issue, still unresolved, is the limited number of cell divisions that human MuSCs can undertake in vitro before losing their myogenic characteristics and/or entering into replicative senescence [7]. To overcome this limitation, along with the scarcity of human material available from DMD muscle, immortalization of MuSCs appears to be a good solution to obtain sufficient material for investigating the molecular mechanisms at work in human dystrophic cells.

Over the past decades, various strategies have been used to bypass the senescence state of human MuSCs. In a first approach, transduction of cells with the viral oncogene simian SV40 large T antigen was able to extend the maximum number of cell divisions by an additional 20 rounds [8]. However, this approach often leads to abnormal chromosomal recombination and to an impaired expression of the myogenic program [8,9,10]. In an alternate approach, ectopic expression of the reverse transcriptase unit of human telomerase (hTERT) further increases the maintenance of MuSCs in vitro, while preserving their differentiation potential. Expression of hTERT was also shown to improve cell survival by counteracting telomere shortening [11,12]. However, the hTERT-expressing MuSCs could not be immortalized due to their propensity to undergo cycle arrest [13]. In an effort to prevent cell cycle arrest, an alternative method was developed that consists of transducing MuSCs with cyclin-dependent kinase 4 (CDK4) or cyclin D1 (CCND1) genes to ensure their proliferative capacity [14,15,16]. Building on these various approaches, the field has adopted a technology whereby the expression of cell cycle drivers CDK4 or CCND1 is combined with that of hTERT to allow the immortalization of human MuSCs that maintain both their differentiation potential and normal karyotype [14,17]. Immortalization of CD56pos purified primary human MuSCs with hTERT and CDK4 transgenes has been performed by several groups [15,18,19]. Transcriptomic analysis showed that immortalized lines retain the myogenic expression patterns of their parent primary populations during myogenic differentiation [19]. Since the development of the immortalization technology, a series of studies have confirmed that this approach can be applied to a variety of neuromuscular diseases, including DMD, myotonic dystrophy 1, congenital muscular dystrophy type 1A, facioscapulohumeral muscular dystrophy, limb-girdle muscular dystrophy type 2B and oculopharyngeal muscular dystrophy [15,18,19,20,21].

In the present study, we used CDK4/TERT expression in primary MuSCs to derive novel MuSC clones from a series of DMD, congenital muscular dystrophies (that can serve as non-DMD myopathic samples) and age-matched control muscles. Several clones were established and characterized for each patient. Previous works have cloned immortalized hMuSCs using low cell density culture and/or a clonal ring [14,15,22], which cannot prevent the presence of cell doublets. Here, we used FACS-derived single-cell cultures to establish several genetically homogeneous immortalized MuSC clones from the same donor, in order to circumvent any impact of the insertion site of the lentiviruses in the genome. All clones were tested to ensure they retained their myogenic characteristics and differentiation potential while ensuring their genomic integrity. Finally, as studies utilizing these cell clones will surely encompass over expression/inhibition of genes of interests, we screened a panel of promoters to identify the preferred gene regulatory unit that should be used in plasmids to ensure expression in these human MuSC clones. The 38 clonal immortalized myogenic cell lines issued from genetically and clinically characterized patients provide a large collection of DMD clones with various genetic defects that are available to the academic community.

## 2. Materials and Methods

### 2.1. Patients

Biopsies were obtained from deltoideus medialis of 8 DMD patients, 4 patients suffering from congenital muscular dystrophies (CMD) and 3 controls (patients showing after investigation no clinical signs of neuromuscular diseases and for whom all the diagnosis workup was normal) used as age-matched control. Biopsies were obtained after institutionally approved protocol and parents or legal representatives gave their written informed consent for the children’s participation to the study (protocol registered at the Ministère de la Recherche and Cochin Hospital Cell Bank, Paris, agreement n°DC-2009-944).

### 2.2. Immortalization of Primary Human MuSCs

After mechanic dissociation of the muscle, enzymatic digestion was performed in DMEM medium (Gibco #10566, Thermofischer Scientific, Waltham, MA, USA) containing 6000 UI/mL Collagenase IA (Sigma-Aldrich#C2674, Merck KGaA, Darmstadr, Germany) for 60–90 min at 37 °C under gentle stirring. Primary cells were cultured and expanded in HAM-F10 medium (Gibco #11550), containing 15% foetal bovine serum (FBS) (Abcys, EurobioScientific, Les Ullis, France, #S1810-500), 100 U/mL penicillin, 100 µg/mL streptomycin (Gibco #15140) and 1% HEPES 1M. Cells were seeded at 5000 cells per cm^2^ in 6-well plates and 6 h later (once the cells are adherent), the medium was replaced by transduction medium (DMEM:M199 medium (Invitrogen #41150, Thermofischer Scientific,) (4:1) containing 10% FBS, 2.5 ng/mL HGF (Sigma-Aldrich, Merck, #H9661), 40 ng/mL Dexamethasone (Sigma-Aldrich #D4902)) containing CDK4 and HTERT lentiviruses. Two lentiviruses for CDK4 and HTERT expression were produced and titrated by the Viral Vectors and Gene Transfer facility at Université René Descartes, Paris. pTrip-CMV-CDK4-DeltaU3 and pTrip-PGK-puromycin-polyA CMV-hTERT variant 1–2 plasmids were used to produce ∆U3 SIN lentiviruses in HEK293T and were purified by ultracentrifugation. Titration was performed in HTC116 cells. Transduction was performed overnight with the 2 lentiviruses to a final concentration of 3 lentiviral particles/cell and medium was replaced by selection medium composed of transduction medium containing 1 µg/mL puromycin (Sigma-Aldrich #P8833). After selection, cells were expanded for about 3 weeks, and supernatants were tested for the presence of HIV-derived p24 using ELISA (INNOTEST, Fujirebio Inc., Courtaboeuf, France), and all the 15 cultures were shown to be negative.

### 2.3. Clonal Isolation and Selection of Clones

After puromycin selection, immortalized human MuSCs (iHMuSCs) were labelled with anti-CD56 APC-conjugated antibodies (BD Pharmingen #555518, BD Biosciences, Franklin Lakes, NJ, USA) and were sorted using a BD FACSAria II to be seeded at one cell per well in 96-well plates. Cells were cultured in growth medium (skeletal muscle basal medium (PromoCell GmbH #C23260, Heidelberg, Germany), containing skeletal muscle supplemental mix (PromoCell #C39365), 10% FBS, 100 U/mL penicillin and 100 µg/mL streptomycin). All along the culture procedure, expression of CD56 was monitored using anti-CD56 APC-conjugated antibodies and a FACSCantoII flow cytometer (BD Biosciences). Growth curves for each clone were built using the number of cells over time of culture. The population doubling was measured as
(1)$$\frac{\log\left(\frac{number\;of\;cells\;cpunted\;at\;T_{1}}{number\;of\;cells\;seeded\;at\;T_{0}}\right)}{\log\left(2\right)}$$

The population doubling time (PDT) was also calculated, using the population doubling over the number of days between T_0_ and T_1_, where T_0_ is the day of seeding and T_1_ the day of the next trypsinization. T_0_ was the time of seeding and T_1_ the time when the culture was stopped.

### 2.4. Assessment of the Myogenic Nature of iHMuSCs

IHMuSCs were seeded at 3000 cells per cm^2^ in 12-well plates containing glass coverslips and were grown in growth medium. After 48 h, cells were washed with phosphate-buffered saline solution (PBS), fixed in formaldehyde 4% for 10 min at room temperature (RT) and were permeabilized in triton X-100 0.1% for 10 min at RT. Immunolabeling was carried out using mouse anti-Pax7 (1:50, Developmental Studies Hybridoma Bank, #PAX7) and rabbit anti-desmin (1:200, Abcam #ab32362) antibodies labelled with donkey anti-mouse IgG and anti-rabbit IgG antibodies (1:200, Jackson ImmunoResearch #715-095-150 and #711-095-152). Nuclei were labeled with Hoechst (Sigma-Aldrich #B2261) for 10 sec at RT, and mounting was performed in Fluoromount (Sigma-Aldrich #F4680).

### 2.5. Two-Dimensional Differentiation Assays

iHMuSCs were seeded at 12,000 cells per cm^2^ in 12-well plates containing glass coverslips and were grown in growth medium for 4 days. Growth medium was replaced by differentiation medium (skeletal muscle basal medium containing 100 µg/mL transferrin (Sigma-Aldrich #T3309), 10 µg/mL human insulin (Sigma-Aldrich #I2643), 100 U/mL penicillin and 100 µg/mL streptomycin) and cells were further incubated for 5 days. Cells were washed with PBS, fixed in formaldehyde 4% for 10 min at RT and were permeabilized in triton X-100 0.1% for 10 min at RT. Immunolabeling was carried out using rabbit anti-desmin (1:200, at 4 °C overnight, Abcam #ab32362, Cambridge, UK) and/or rabbit anti-dystrophin (1:50, at 4 °C overnight, Abcam #15277) antibodies revealed with donkey anti-rabbit IgG and/or anti-mouse IgG antibodies. Nuclei were labeled with Hoechst for 10 sec at RT, and mounting was performed in Fluoromount.

### 2.6. Three-Dimensional Differentiation Assays

Twelve-well plates were coated with 500 µL of diluted Matrigel (Matrigel:Growth medium [1*v*:9*v*]) for 30 min at 37 °C. Wells were washed twice with PBS, and iHMuSCs were seeded at 30,000 cells per cm^2^ in growth medium. After 6 h of incubation, medium was replaced with differentiation medium. One day later, medium was removed and cells were covered with a thin layer of diluted Matrigel without serum (Matrigel:skeletal muscle basal medium without serum [1*v*:2*v*]). Covered cells were incubated for 30 min at 37 °C, and following Matrigel solidification, 500 µL of differentiation medium were added and cells were further incubated for 4 days. Cells were washed twice with PBS, fixed in formaldehyde 4% for 10 min at RT and permeabilized in triton X-100 0.1% at RT. Immunolabeling was performed using mouse anti-myosin heavy chain (MHC) antibodies (1:100, Developmental Studies Hybridoma Bank #MF20, Iowa city, IA, USA) revealed with donkey anti-rabbit IgG antibodies. Phalloidin conjugated with Atto 488 (Sigma #49409) was added to label actin. Nuclei were labeled with Hoechst for 10 sec at RT, and cells were mounted in Fluoromount.

### 2.7. Propidium Iodide Staining

iHMuSCs were seeded in proliferation condition at 3000 cells per cm2 for 3 days. Cells were trypsinated, washed with PBS and fixed with 66% of cold ethanol overnight at −20°C. Fixed cells were washed with PBS and stained for 30 min with 50 µg/mL propidium iodide (Sigma-Aldrich #P4864) and 10 µg/mL RNAse A (ThermoFisher Scientific #12091-021). Cells were washed with PBS and were analyzed using a FACSCantoII flow cytometer.

### 2.8. Plasmid Construction

The pLenti-GIII-CMV-mCherry plasmid was built by insertion of a PCR-amplified mCherry gene in pLenti-GIII-CMVplasmid (ABMgood #LV587), digested at cutting sites NheI-XhoI, using a T4 ligase (Invitrogen #15224025). The pLenti-GIII-EF1α-mCherry, pLenti-GIII-PGK-mCherry and pLenti-GIII-EF1a-mCherry plasmids were built by insertion of PCR amplified EF1α, PGK or UbC promoters in the previously built pLenti-GIII-CMV-mCherry plasmid, digested at either AvrII and NheI cutting sites for EF1α or cutting sites ApaI and BamHI for PGK and UbC, using a CloneEZ^®^ PCR Cloning Kit (GenScript #L00339) and following the manufacturer’s instructions. The PCR amplification of mCherry, EF1α, PGK and pUb were performed with Phusion^®^ High-Fidelity DNA Polymerase (NEB #M0530S) on pN1-mCherry (Clontech, #632523), pLenti-EF1α (ABMgood #LV588), PGK-DTA-BpA 23 and Ul4-GFP-SIBR 24 plasmids respectively and following the manufacturer’s protocol. Primers used for PCR amplification are listed in Table 1.

### 2.9. Test of Lentiviral Promoters

Lentiviral production was carried out by CaCl2 transfection of HEK293T cells with 3 plasmids. Transfection was performed with 1 × 10^6^ HEK293T cells using 19.9 µg of constructed lentiviral vector or a pLenti-GIII-EF1α empty (ABMgood #LV588), 5.93 µg of MD2.G plasmid containing VSV-G envelope gene under the control of a CMV promoter (Addgene #12259) and 14.88 µg of psPax2 packaging plasmid containing HIV-gag and HIV-polymerase gene under the control of a CMV promoter (Addgene #12260) over 15 h. The supernatant containing lentivirus was collected 24 and 48 h after the end of transfection, filtered and concentrated using sucrose buffer at low centrifugation as described in [23]. Lentiviral titration was estimated by transfection of HEK293T cells with concentrated lentivirus in DMEM high glucose medium (Gibco #1196044) containing 10% FBS, 100 U/mL penicillin and 100 µg/mL streptomycin and supplemented by 6 µg/mL polybren (Sigma-Aldrich #107689). One day after transfection, cells were selected using growth medium supplemented with 2 µg/mL of puromycin. Multiplicity of infection (MOI) was calculated by counting the remaining cells 3 days after the start of selection. After transduction tests with different MOI ranges from 1 to 10 particles/cell, the more efficient concentration was defined as a MOI of 4 particles/cell. iHMuSCs were seeded at 3000 cells per cm^2^ with growth medium in 6-well plates. Six hours later, adherent cells were incubated with growth medium supplemented with 6 µg/mL polybren and concentrated lentivirus to a final MOI of 4 lentiviral particles/cell, based on the lentivirus infection efficacy on C2C12 cells (not shown). Medium was replaced by fresh growth medium 36 h later and cells were trypsinated 1 and 5 days later for flow cytometry analysis of the cherry fluorescence using a FACSCantoII flow cytometer. In parallel, cells were cultured on glass coverslips for 1 and 7 days of culture, cells were washed with PBS and fixed with formaldehyde 4% for 10 min at RT. Immunolabeling was carried out using primary rabbit anti-desmin antibodies as described above.

### 2.10. Quantitative RT-PCR

Total RNAs were extracted from growing iHMuSCs using Trizol (Invitrogen, #15596026) following the manufacturer’s protocol. Controls were 2 non-immortalized primary HMuSCs. Each sample was tested in triplicate. Cultured cells were directly treated with Trizol. One microgram of total RNA was reverse-transcripted using Superscript II Reverse Transcriptase (ThermoFisher #18064022) in 20 µL at 42 °C for 50 min prior to being diluted 10 times. Quality of RNA was checked using Nanodrop. The quantitative RT-PCR were performed using a CFX96 Real-Time PCR Detection System (Bio Rad). The 10 µL final volume of reactive mixture contained 2 µL of diluted cDNA, 0.5 µL of primer mixture, 2.5 µL of water and 5 µL of LightCycler 480 SYBR Green I Master Kit (Invitrogen, Thermofisher Scientific #04707518001). After initial denaturation of 2 min, the amplification was performed with 45 cycles of 95 °C for 10 s, 60 °C for 5 s and 72 °C for 10 s. The calculation of calibrated, normalized relative quantity (CNRQ) was performed using AP3D1 housekeeping gene (there is no variation of expression during myogenesis [24]), with an inter-run calibrator sample. The genes analyzed using quantitative RT-PCR were TERT and CDK4. Primers used for quantitative PCR are listed in Table 1.

## 3. Results and Discussion

### 3.1. Immortalized Myogenic Cell Lines Generated from Duchenne Muscular Dystrophy, Congenital Muscular Dystrophies and Control Muscles 

We identified 8 DMD patients and 4 patients suffering from congenital muscular dystrophies (CMD) for which genetics and clinic characteristics have been established and 3 age-matched controls with normal muscle histology (Table 2). CMD included mutations in collagen VI, Lamin A/C, Laminin alpha-2 genes and an unknown cause. The genetic causes of DMD for the 8 patients are indicated in Table 2. Data collected for the donors included creatine kinase (CK) level at the time of biopsy collection, age at first walking, the presence or absence of cardiomyopathy and scoliosis and whether patients underwent scoliosis surgery (Table 2).

MuSCs were isolated as CD56pos cells from the 15 patients, expanded, infected with lentiviruses expressing CDK4 and HTERT and selected using puromycin as previously described [20]. CDK4 overexpression blocks the p16INK4a-Rb pathway while hTERT maintains telomere length, allowing MuSC immortalization [17,20,22]. The bulk cell populations were first selected using puromycine, removing cells lacking the *HTERT* transgene. Then, cells were expanded, in order to deplete cells lacking the *CDK4* transgene [13].

Infection of myoblasts with lentiviruses is expected to have generated cells that integrate variable copy numbers of the transgenes into different genomic loci. This is likely to cause high intercellular variability and heterogeneous cell populations. To select clones presenting a homogeneous phenotype and genotype, we carried out FACS single cell sorting of CD56pos immortalized cells and amplified these clonal cultures, referred to as iHMuSCs for immortalized human muscle stem cells. These clones were then analyzed for their expansion capacity, myogenic nature and myogenic differentiation potential.

### 3.2. Selection of iHMuSCs Exhibiting Efficient Growth Capacity

Expanding clones were first tested for their capacity to proliferate. Basically, two types of clones were observed: clones that were not capable of expansion after a few weeks, and that were discarded from further analyses, and clones that expanded efficiently and were selected. Figure 1A shows examples of clones that replicated rapidly from the time of seeding, exhibited a regular growth and showed population doubling times ranging from 2.5 to 5.4 days in growing conditions. While some variability in population doubling time was observed, no significant difference was identified when considering the pathology, i.e., controls versus DMD versus CMD (Figure 1A,C). Moreover, variability in population doubling time was observed between clones issued from the same patient, as exemplified for two patients in Figure 1B. Distribution of the population doubling time for all the selected clones is shown in Figure 1D. Therefore, the proliferative capacity was characteristic of each clone and may be related to the sites of insertion of CDK4 and HTERT genes in the genome. Thus, several clones were generated from each patient in order to allow future investigators to work on several clones from the same patient to avoid potential bias induced by the site of insertion of the lentiviral-driven genes.

Both control and DMD iHMuSC clones were selected for their efficient growth. The proliferation capacity of MuSCs in DMD has been a matter of debate. Earlier works reported a defect in both proliferation and differentiation of the DMD myoblasts [25,26,27,28] and others not [29], but at that time there was no method of purification of cell cultures, which contained non-myogenic cells. Later, it was shown that pure myogenic stem cells from human DMD muscle do not show alteration in their proliferative capacity as compared with cells issued from healthy muscle [20,30].

### 3.3. Myogenic Nature of iHMuSCs

We confirmed that CDK4 and TERT transduction was efficient, through RT-qPCR of CDK4 and TERT genes in growing iHMuSCs, as compared with primary HMuSCs. The latter, issued from two healthy donors, exhibited a very low and no expression of CDK4 and TERT genes, respectively. In iHMuSCs, CDK4 expression was 3 to 35 fold higher and the expression of hTERT was triggered (minimum 1300 fold) (Figure 2A,B), confirming that the clonal cultures expanded from transduced cells.

The myogenic nature of iHMuSCs was investigated via Pax7 and desmin immunostaining on three healthy and three DMD growing iHMuSC clones issued from various donors (Figure 2C). In all six clones tested, 100% of iHMuSCs expressed both desmin and Pax7 protein, confirming that they were myogenic cells. Moreover, all 38 clones stemmed from CD56 expressing cells and then were further tested after expansion for their expression of CD56, which is a canonical marker of MuSCs in human, where cells that express CD56 invariably express Pax7 [31,32]. Additionally, we investigated the expression of dystrophin in differentiated iHMuSC clones, since in mouse, dystrophin expression is reported to be specific of satellite cells and differentiated myoblasts, but absent from growing myoblasts [33]. As expected in iHMuSCs derived from control MuSCs, differentiated myotubes expressed both the cytoplasmic and nuclear isoforms of dystrophin [34], using an antibody targeting all five isoforms (epitope between exons 77 to 79) (Figure 2D). Inversely, DMD-derived iHMuSCs displayed only the nuclear isoforms (Figure 2D), confirming the lack of muscle specific Dp427m dystrophin expression. 

### 3.4. Selection of iHMuSCs Exhibiting Efficient Myogenic Differentiation

The above selected growing clones were further checked for their ability to sustain high-level expression of CD56, a specific marker of human myogenic cells [6,31,35,36]. Only clones exhibiting CD56 expression in more than 90% of the cells were retained. Previous studies showed that in bulk transduced cultures, CDK4 and HTERT overexpression triggers MuSC immortalization without interfering with their myogenicity [14,37]. We implemented a complete analysis of the differentiation capacities of each of the selected clones. Myogenic differentiation was classically triggered by culturing iHMuSCs in differentiation medium, containing low mitogen concentration and supplemented by insulin and transferrin. Myogenic differentiation capacity of the cells was assessed using immunofluorescence using desmin staining to visualize the formation of multinucleated myotubes. The formation of myotubes was estimated in a semi-quantitative manner to exclude clones showing no or poor ability to differentiate. Figure 3 (2D-glass) shows examples of iHMuSCs forming myotubes on a 2D glass culture support. We also performed a 3D culture that improves myogenic differentiation, by culturing the cells between two thin layers of Matrigel, which is a mixture of matrix proteins constitutive of basal lamina that promotes muscle cell growth and differentiation [38]. Figure 3 (3D-matrigel) shows that iHMuSCs from either control, CMD or DMD patients formed large elongated myotubes containing numerous nuclei and having developed their contractile apparatus, as assessed by MHC expression.

In total, the selection process provided 38 iHMuSC clones exhibiting efficient proliferative capacity, sustained CD56 expression and showing excellent capacities for myogenic differentiation, available for 15 genetically and clinically characterized patients (Table 3).

### 3.5. Characteristics of Long-Term Cultured iHMuSCs

Eight clones were tested for long-term culture, up to 240 days. For all clones, regular growth was observed over time, allowing cells to perform between 60–100 population doublings after 240 days (Figure 4). CD56pos expression was maintained at a high level since all but one clone showed greater than 93% of the cell expressing the myogenic membrane protein over long-term cultures (with clone DMD7-E3 showing 82.9% of CD56pos cells) (Table 3). As for younger cultures, high CD56 expression was associated with the capacity to differentiate and to form myotubes in all clones tested but one (Table 3). It was previously shown that SV40 large T gene overexpression leads to abnormal chromosomic recombination in myoblasts and to an impaired expression of the myogenic program [8,9,10]. To estimate potential major changes in chromosomic recombination and in cell cycle, we performed cell cycle analysis using FACS with propidium iodide. Four clones were tested at early and old population doublings. Figure 5 shows that all clones presented a normal cell cycle profile during short-term cultures (left panel) that was maintained in long-term cultures (right panel). No mark of polyploidy was observed. We thus assumed that no major chromosomal abnormalities appeared in iHMuSCs, although one has to keep in mind that mutations and short chromosomal modifications may occur in cells kept for a long time in culture.

### 3.6. Test of Promoter Activity for Lentiviral Transduction in iHMuSCs

A potential application of iHMuSCs is gene overexpression or silencing by gene editing. To maintain either stable gene overexpression or a stable genome edition, it is necessary to carry out a lentiviral infection [39]. However, promoters may have various transcriptional efficiency depending on the cell type [40]. In order to identify promoters that are efficient for further stable gene expression in iHMuSCs, we performed lentiviral transduction with vectors expressing the mCherry reporter gene under the control of various promoters in control iHMuSCS, here considered a cell type. We tested human cytomegalovirus promoter (CMV), human elongation factor 1α promoter (EF1α), phosphoglycerate kinase promoter (PGK) and ubiquitin C promoter (UbC). Figure 6A shows that PGK and UbC promoters triggered very low level expression of mCherry in iHMuSCs, a result confirmed using flow cytometry (17.1 and 38.1% of positive cells for PGK and 18.5 and 44.3% of positive cells for UbC, at 1 and 5 days after transfection, respectively) (Figure 6C). CMV and EF1α were the most active promoters since 90.4 and 64.2% of cells were positive for mCherry expression 24 h after transduction (Figure 6A,C). However, at 5 days, the number of positive cells had declined by 10% when the promoter was driven by the CMV promoter (Figure 6C), such a decrease over time being reported in other models [41,42]. Inversely, almost no decrease (2.9%) was observed using EF1α promoter to drive mCherry expression after 5 days and an increase was noticed for PGK and UbC (21 and 26.2%, respectively) (Figure 6B,C). Thus, to alter gene expression in iHMuSCs, the CMV and EF1α promoters induced fast and strong expression of the transgene, while the PGK and UbC promoters ensured a later expression. CMV or EF1α should be preferred to obtain fast and stable expression of a transgene. In both cases, cell selection is recommended to work with cell populations expressing the transgene homogeneously in order to maintain a stable expression.

## 4. Conclusions

In the present study, we used the well-characterized immortalization procedure of primary human MuSCs [19,20], and we performed single cell-derived cultures to establish clones that present homogeneous genetics and cell behavior. This study provides the description of 38 clones of iHMuSCs coming from 12 patients, and including 6 clones from 3 control muscles, 12 clones from 4 CMD patients and 20 clones for 8 DMD patients, who exhibited a large variety of genetic defects. Each patient was genetically and clinically characterized, and each clone was validated for its myogenic nature (Pax7 and desmin expression), proliferative capacity, maintenance of CD56 expression and ability to differentiate/fuse. These clones provide a useful material for investigation of different aspects of DMD biology such as metabolism, gene expression, chromatin organization and membrane changes or for screening of therapeutic strategies in complement with the known animal models. The information for each patient allows choice of the clones depending on clinical features, and patient genotyping will be essential for the design of exon skipping strategies such as the recent experiments using CRISPR-Cas9ti edit the *DMD* gene [43,44]. Moreover, working with several clones from the same patients avoids bias due to the insertion sites of the immortalization genes. Although providing an unique material for the investigation of molecular alterations in DMD, one has to keep in mind that iHMuSCs have a slightly modified genome caused by the insertion of CDK4 and HTERT genes that may induce off-target changes depending on the insertion sites of the transgenes. Thus, it is recommended that several clones from the same parental transduced culture be used in a given experimental setup to exclude any effect due to the site of insertion of the transgenes or clone specificities.

Here, iHMuSCs were produced from DMD patients carrying eight different genetic alterations allowing the investigation of DMD mechanisms independently of—or dependent upon—a specific mutation. Similarly, iHMuSCs were produced from CMD patients of various origins allowing investigation of dystrophic cells unrelated to dystrophin loss. The 38 iHMuSCs clones will be a valuable resource for the academic scientific community and will be made available through signed material transfer agreements.

## Figures and Tables

**Figure 1 cells-09-01780-f001:**
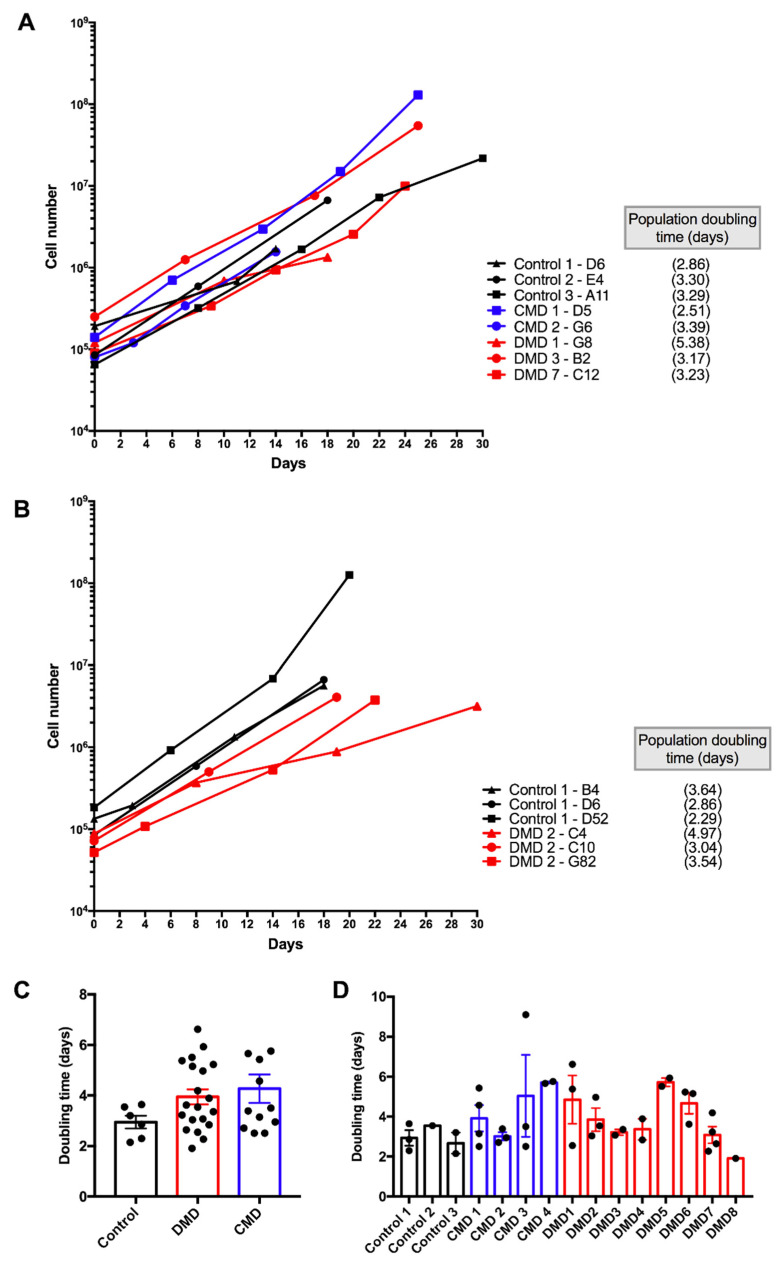
Growth curve of immortalized human muscle stem cell (iHMuSC) clones. IHMuSC clones were expanded in growing medium. (**A**) Growth of eight clones from eight different patients. (**B**) Growth of three different clones from the same patient (one control and one DMD patient are shown). (**C**) Population doubling time of control, DMD (Duchenne Muscular Dystrophy) and CMD (Congenital Muscular Dystrophy) derived iHMuSCs. (**D**) Comparison of population doubling time from each donor (clones from the same donor are gathered in one bar). Statistical analyses were done using one-way ANOVA.

**Figure 2 cells-09-01780-f002:**
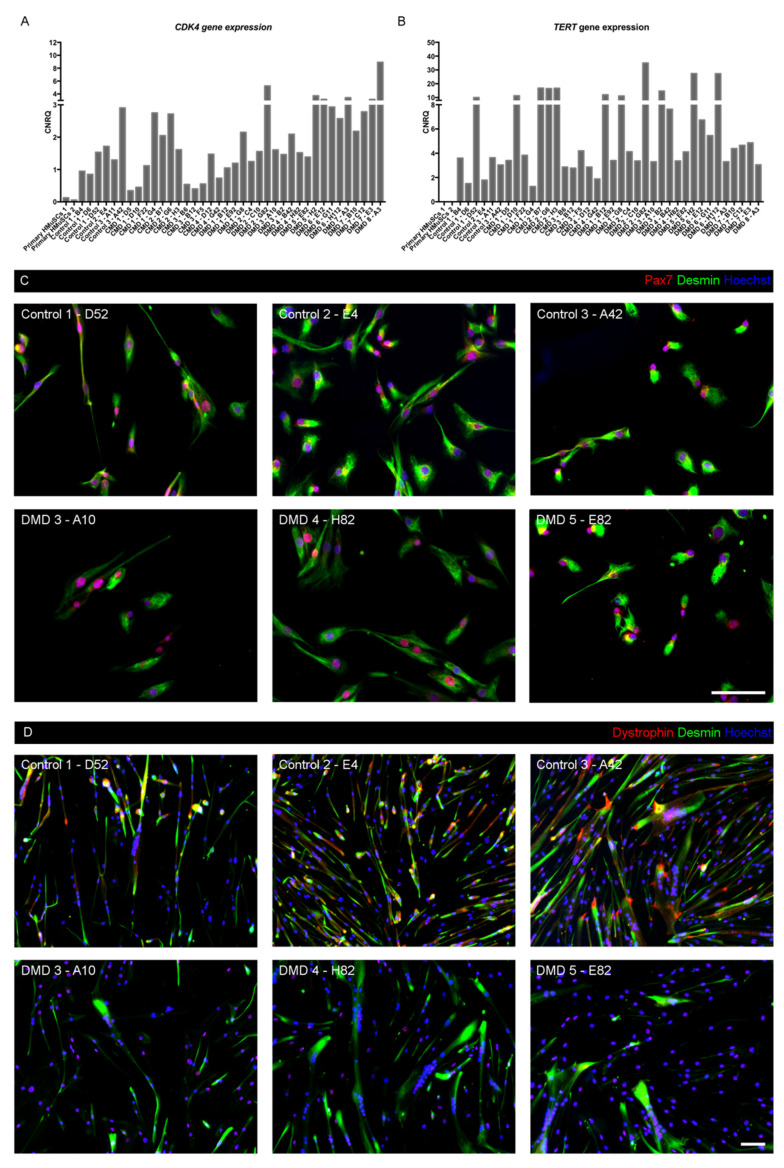
Myogenic nature of iHMuSC clones. (**A**,**B**) Primary HMuSCs and iHMuSC clones were tested for CDK4 (**A**) and TERT (**B**) gene expression using RT-qPCR. (**C**) Immunostaining for the myogenic markers Pax7 (Red) and desmin (green) in growing iHMuSC clones. (**D**) Immunostaining desmin (green) and dystrophin (red) in differentiated iHMuSC clones. Nuclei are labelled with Hoechst (blue). Bars = 100 µm.

**Figure 3 cells-09-01780-f003:**
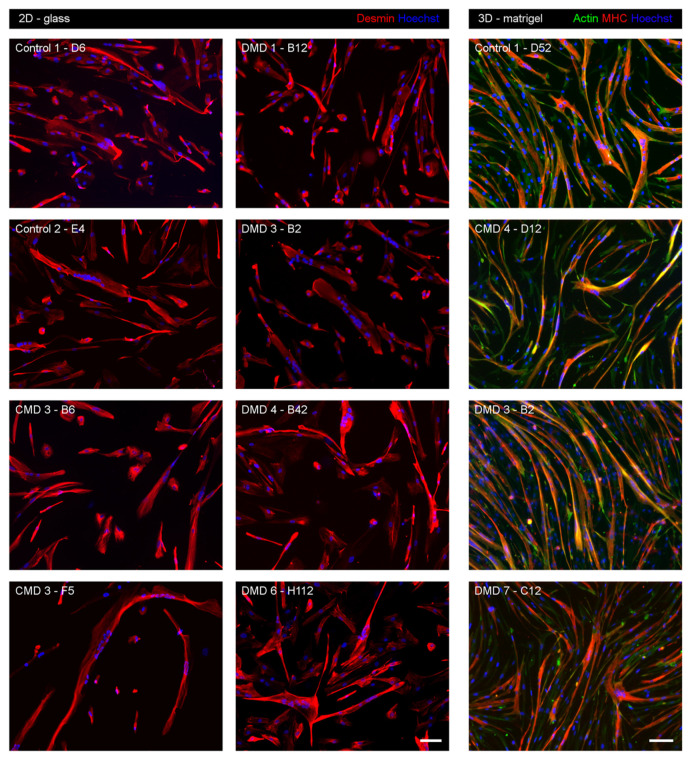
Myogenic differentiation of iHMuSC clones. Clones selected for their good growth capacity were tested for their myogenic differentiation capacity. In the 2D glass condition (left and middle panels), clones were differentiated on glass culture supports for 5 days before the detection of desmin (red) using immunofluorescence. In the 3D-Matrigel condition (right panel), clones were differentiated between 2 thin Matrigel coats for 5 days before the detection of actin (green) and MHC (red) using immunofluorescence. Nuclei are labelled with Hoechst (blue). Bars = 100 µm.

**Figure 4 cells-09-01780-f004:**
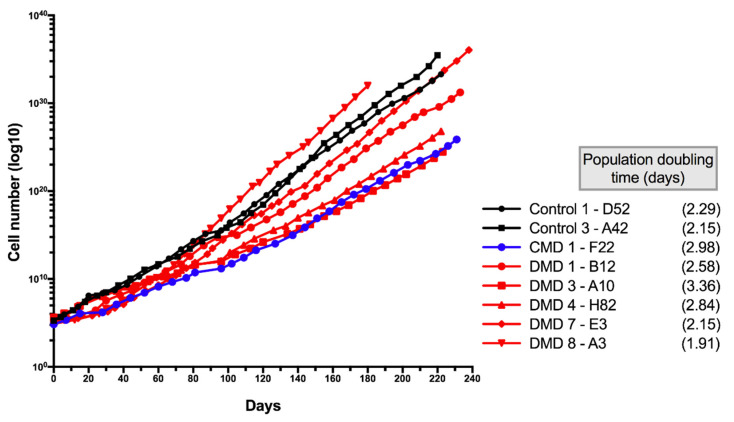
Long-term culture of iHMuSC clones. Eight clones from eight patients were cultured up to 240 days.

**Figure 5 cells-09-01780-f005:**
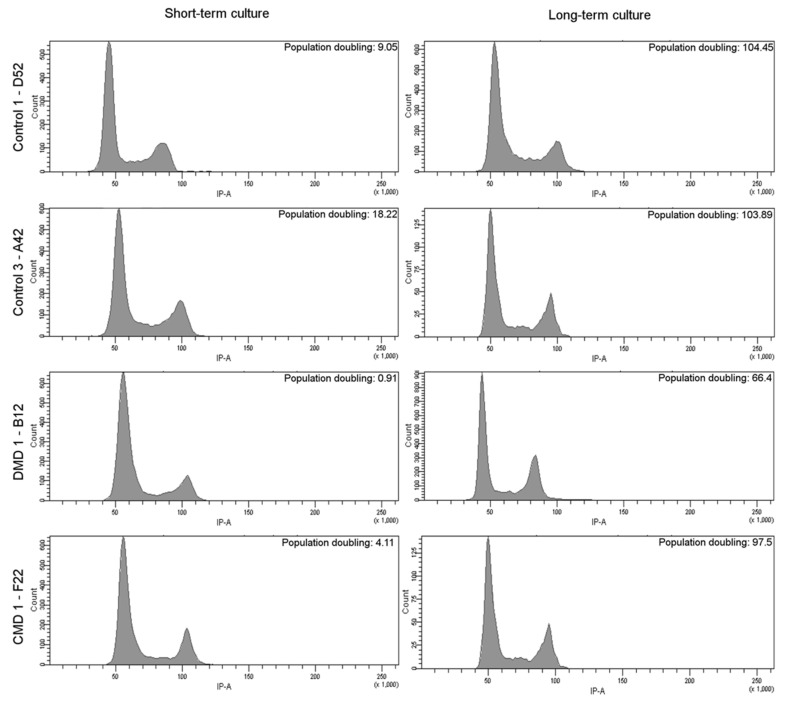
Cell cycle analysis of short-term and long-term cultured iHMuSC clones. Four randomly chosen clones were tested for cell cycle analysis at short- (left panel) and long- (right panel) term culture. Flow cytometer analysis of propidium iodide staining is shown, reflecting the amount of DNA in the cells.

**Figure 6 cells-09-01780-f006:**
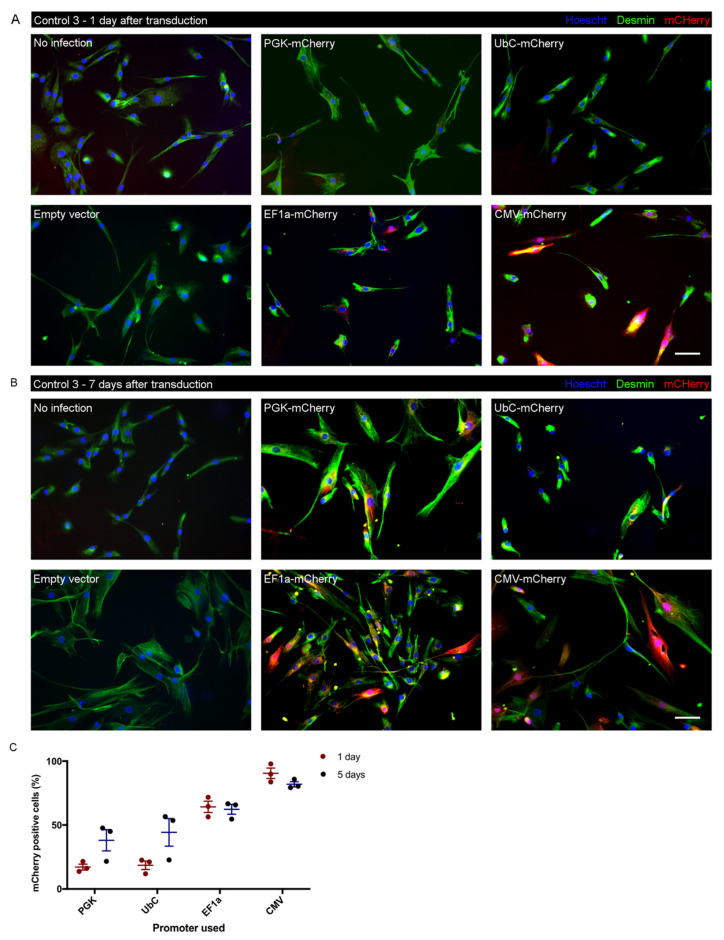
Lentiviral transduction of mCherry reporter gene under the control of several promoters. iHMuSCs control 1—D52, control 2—E4 and control 3—A42 were transduced with lentiviruses containing mCherry under the control of PGK, UbC, EF1 or CMV promoters or with an empty lentivirus. (**A**,**B**) Immunostaining was performed 1 (**A**) and 7 (**B**) days after transduction for the detection of desmin (green). mCherry is red. Nuclei are labelled with Hoechst (blue). Bars = 100 µm. (**C**) Quantification of the expression of mCherry in iHMuSCs using flow cytometry 1 and 5 days after the lentiviral transduction. Results are given in percentage of mCherry positive cells and were done on the 3 different clones: control 1—D52, control 2—E4 and control 3—A42 (means ± SEM).

**Table 1 cells-09-01780-t001:** Primers used for PCR amplification.

Primer Name	Sequence	Anneling Temperature
mCherry NheI Forward	GAGATCGCTAGCGGGCCCGCCACCATGGTGAGCAAGGGCGAGGAG	60.9 °C
mCherry XhoI CMV Reverse	GAGATCCTCGAGCTACTTGTACAGCTCGTCCATG	60.9 °C
Promoter EF1ɑ Forward	TCGGGTTTTTCGAACGGCTCCGGTGCCCGTCAG	60 °C
Promoter EF1ɑ Reverse	ATGGTGGCGGGCCCGTCACGACACCTGAAATGGAAG	60 °C
Promoter PGK/UBC Forward	GATCTCGACGGTATCGAAAGC	60 °C
Promoter PGK Reverse	CCATGGTGGCGGGCCCGAATTCGATCTCGGATCCGAAAGCGAAGGAGCAAAGCTG	60 °C
Promoter UbC Reverse	CCATGGTGGCGGGCCCGAATTCGATCTCGGATCCGTCTAACAAAAAAGCCAAAAAC	60 °C
AP3D1 Forward	GGCATCCGTAACCACAAGGA	60 °C
AP3D1 Reverse	TTGTCCTGCTTCAGCTCCTG	60 °C
CDK4 Forward	GGCTGAAATTGGTGTCGGTG	60 °C
CDK4 Reverse	CACGAACTGTGCTGATGGGA	60 °C
TERT Forward	TGTACTTTGTCAAGGTGGATGTGA	60 °C
TERT Reverse	GCTGGAGGTCTGTCAAGGTAGAG	60 °C

**Table 2 cells-09-01780-t002:** Clinical data of patients.

Patient	Sex (M/F)	Age at the Time of Biopsy (m)	Diagnosis	Genetic Results	CK Blood Level at Diagnosis (IU/L)	Age at First Walking (m)	Cardiomyopathy Age at Onset (m)	ScoliosisAge at Onset (m)	Scoliosis Surgery/Age at Surgery
Control 1	M	41	control	/	nl	24	N	N	N
Control 2	M	14	control	/	nl	24	N	N	N
Control 3	M	19	control	/	nl	18	N	N	N
CMD 1	M	144	Collagen VI related myopathy	C6210 + 5G > A	nd	72	N	Y	Y/145
CMD 2	F	144	CMD	genetic unknown causes	nl	18	N	Y/195	Y/231
CMD 3	F	39	Laminopathy LMNA	c.94–96 deletion; p.lys32 deletion	860	never walking	N	N	N
CMD 4	F	12	LAMA2 related myopathy	c.1553 deletion GTT; pCys518 deletion and c.2866 deletion T	14400	never walking	N	N	N
DMD 1	M	54	DMD	c.3–26 duplication	18000	24	N	N	N
DMD 2	M	87	DMD	c.8–43 deletion	nd	13	Y/109	Y/141	Y/148
DMD 3	M	97	DMD	c.433 C > T substitution (p.R145X)	12881	15	N	Y/161	Y/161
DMD 4	M	25	DMD	c.8562 deletion A; p.Glu2854Asp fs X2	19000	24	N	N	N
DMD 5	M	79	DMD	c.5758 C > T substitution; p.Gln1920X	8041	15	N	N	N
DMD 6	M	nd	DMD	Exon skipping 19 / IVS 19 +1 G > C/c.2380 + 1 G >C	nd	nd	Y/82	N	N
DMD 7	M	89	DMD	c.50–59 dup	47270	15	N	N	N
DMD 8	M	nd	DMD	nd	nd	nd	nd	nd	nd

CMD, congenital muscular dystrophy; DMD, Duchenne muscular dystrophy; M, male; F, female; CK, creatin kinase; IU/L, international unit per liter; nl, normal; Y, yes; N, no; m, month; nd, non-determined.

**Table 3 cells-09-01780-t003:** Summary of characterized immortalized human myogenic stem cell clones.

Patients	Clones	Population Doublings ^a^	CD56 ^pos^ Cells (%)	Differentiation	Doubling Time (days) ^b^
Control 1	B4	3.33	99.50	++	3.64
	D6	2.80	99.80	+++	2.86
	D52 *long term*	5.22	99.70	+++	2.3
		*73.23*	*99.90*	*++*	
Control 2	E4	4.69	99.40	++	3.55
Control 3	A11	1.82	98.07	++	3.2
	A42 *long term*	3.87	98.20	++	2.15
		*73.83*	*99.90*	*+++*	
CMD 1	D5	6.50	99.00	+++	2.51
	D10	8.73	98.90	++	5.43
	F22 *long term*	4.53	98.50	+++	3.15
		*69.86*	*99.80*	*+++*	
	G4	5.15	98.91	+++	4.57
CMD 2	B7	4.74	99.80	+++	2.95
	G6	4.29	99.80	+++	3.39
	H3	5.80	99.70	+++	2.71
CMD 3	B6	4.42	96.00	++	3.5
	B12	2.00	97.30	+++	2.51
	F5	1.96	90.00	+++	9.1
CMD 4	D12	5.40	93.30	+++	5.76
	G42	2.51	91.00	+++	5.66
DMD 1	B12 *long term*	5.17	99.00	++	2.55
		*66.39*	*93.40*	*++*	
	E92	3.39	99.00	++	6.62
	G8	3.48	97.20	+++	5.38
DMD 2	C4	2.07	86.00	+++	4.97
	C10	5.82	93.10	++	3.04
	G82	6.18	98.50	++	3.54
aDMD 3	A10 *long term*	3.90	99.30	++	3.36
		*42.39*	*95.10*	*+*	
	B2	7.77	89.00	+++	3.07
DMD 4	B42	3.61	98.10	++	3.89
	H82 *long term*	5.39	99.80	++	2.84
		*52.30*	*nd*	*+++*	
DMD 5	E82	4.32	98.00	++	5.93
	H2	2.56	100.00	++	5.51
DMD 6	E12	2.65	96.70	+++	3.62
	G11	6.45	93.20	+++	5.23
	H112	3.39	91.70	++	5.15
DMD 7	A3	5.80	93.70	+++	2.64
	B10	8.40	98.00	+++	4.19
	C12	6.80	99.30	+++	3.23
	E3 *long term*	15.96	99.00	++	2.27
		*119.30*	*82.90*	*nd*	
DMD 8	A3 *long term*	5.72	99.20	++	1.91
		*63.55*	*98.00*	*nd*	

In italic, long term culture; +, poor fusion; ++, small myotubes; +++, large myotubes; nd, non-determined. ^a^ Population doubling of the culture at the time of analysis of CD56 expression and of differentiation. ^b^ Doubling time along the culture from the initial seeding until the indicated population doubling.

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
