# Peer review of "Derivation and Characterization of Immortalized Human Muscle Satellite Cell Clones from Muscular Dystrophy Patients and Healthy Individuals"

_cells, 2020, doi:10.3390/cells9081780_

Round 1
Reviewer 1 Report
In this study, the authors generated 38 immortalized myogenic cell lines from primary muscle stem cells (MuSCs) of Duchenne muscular dystrophy (DMD) (8 patients, 20 clones), congenital muscular dystrophy (CMD) (4 patients, 12 clones) and healthy control (3 individuals, 6 clones). In most of the immortalized cell lines, nearly 99% of the cells are CD56-positive even after expansion and all clones express Pax7. All clones keep high proliferative activity and well differentiate into myotubes upon induction. They are undoubtfully useful for investigation on pathogenesis and therapy of muscular dystrophies. Many researchers would request them the clones. On the other hand, immortalized human cell lines from patients with neuromuscular disorders (CMD, DMD, FSHD, LGMD2B and OPMD) using CDK4/hTERT has been already reported in a previous study (Mamchaoui et al., Skeletal Muscle, 2011, 1:34). In addition, the manuscript shows no pathological features of each cell line. The novelty of this study is limited.
Major comments:
・Why the author call their clones “stem cell clones”? Percentage of PAX7-positive cells of each clone would be informative.
・Figure 6C: Practically, which promoter is suitable for the stable expression in immortalized human muscle stem cell clones is important information. Only five days culture is not enough to evaluate the usefulness of 5 different promoters. mCherry expression after longer culture period would be more helpful.
Minor comments:
・Figure 1C. DMC should be CMD. Figure1D: Statistical analysis is not clear.
・Line 378: They should write “12 patients” because three of them are healthy controls.
Author Response
Major comments:
・Why the author call their clones “stem cell clones”? Percentage of PAX7-positive cells of each clone would be informative.
After immortalization of the primary myogenic cell cultures with lentiviruses, the immortalized cell populations were sorted for their expression of CD56, were seeded as one cell per well, and were further expanded. Therefore these expanded cell populations are clones because they stem from a unique cell; they all carry the same genetic modifications with the insertion of CDK4 and hTERT transgenes at the same loci. However to avoid any confusion, we propose to modify the title of the manuscrit as follows: Derivation and characterization of immortalized human muscle satellite cell clones from Muscular Dystrophy patients and healthy individuals. We wish to keep the MuSC abbreviation since it is now widely used in the field.
Concerning the expression of Pax7. Figure 2C shows that 100% of the cells in the clonal population expressed Pax7. This was specifically shown for 6 clones. However, all the 38 clones were tested for their expression of CD56 (Table 3), which is an unequivocal marker of MuSCs in human, all cells expressing CD56 also express Pax7 (Barruet et al. Functionally heterogeneous human satellite cells identified by single cell RNA sequencing. Elife. 2020 9:e51576; Xu et al. Human Satellite Cell Transplantation and Regeneration from Diverse Skeletal Muscles. Stem Cell Reports. 2015 5:419-434). We therefore assume that the percentage of CD56pos cells reflects the myogenicity of the clonal cell populations. Precisions have been added in the result sections, lines 293-5.
・Figure 6C: Practically, which promoter is suitable for the stable expression in immortalized human muscle stem cell clones is important information. Only five days culture is not enough to evaluate the usefulness of 5 different promoters. mCherry expression after longer culture period would be more helpful.
After the test of the 4 promoters we showed that the use of either EF1a or CMV should be preferred in state of PGK or UbC which presented a delay and low expression in iHMuSCs. But even if these promoters present a high transgene expression, it is necessary to select a homogenous population to maintain a stable gene expression.
We don’t have longer culture time with the mCherry reporter gene. However in another work, we tested a 5 week-long culture after transduction with CMV promoter and puromycin selection in primary MuSCs and we saw a decreased activity of CMV leading to a low transgene expression. Because previous works already suggest that CMV activity declines with time, we suggested that the decreased mCherry expression at 5 days is the beginning of this loss of activity. In this case, to maintain a stable transgene expression, it is necessary to select the clones. The clarifications have been added in the result section, lines 383-8.
Minor comments:
・Figure 1C. DMC should be CMD.
This was corrected
・Figure1D: Statistical analysis is not clear.
In Figures 1C and 1D, the doubling time of each cell population was compared.
Results for One-Way Anova test in Figure 1C are:
|
ANOVA summary |
|
|
F |
1.588 |
|
P value |
0.2187 |
|
P value summary |
ns |
|
Significant diff. among means (P < 0.05)? |
No |
|
R square |
0.0832 |
Results for One-Way Anova test in Figure 1C are:
|
ANOVA summary |
|
|
F |
1.316 |
|
P value |
0.2713 |
|
P value summary |
ns |
|
Significant diff. among means (P < 0.05)? |
No |
|
R square |
0.4447 |
We also rephrased the legend for a better understanding of what the bars represent in Figure 1D.
・Line 378: They should write “12 patients” because three of them are healthy controls.
This was corrected
Reviewer 2 Report
The work submitted by Massenet et al presents the characterization of immortalized clones obtained from human muscle stem cells from Duchenne Muscular Dystrophy patients and controls. The study is divided into 2 part: first, the establishment and characterization of 38 immortalized clones from 8 DMD patients, 4 CMD patients and 3 controls; second, the choice of the best promoter for gene expression in such a model.
The work is huge, seriously designed and rigorous. However, I have a major concern regarding the second part and its relevance.
For qPCR experiments, how was the normalized gene chosen? How were the MIQE guidelines followed?
hTert and CDK4 were analysed by qRT-PCR. How did the author made the difference between a transcript and a potential plasmid DNA contamination that is still present in the infectious supernatant even several days after transduction?
The authors mention the use of 2 lentiviruses for immortalization but only mention 1 resistant drug (Puromycin) to select their population. How can they be sure that the population is pure and immortalized by selecting a single gene?
Could the authors detailed the plasmids and the detailed protocol used for immortalization knowing that it is not described in the citation they mentioned ?
What are the characteristics of the hMuSCs native population in terms of doubling the population in the short and long term?
In figure 1, C and D, how is the doubling time generated? Is it an average of the total doubling rate calculated over the life span? If yes, how long was the life span for each clones? Because, as shown in figure 1A, some clones were stopped at 14 days and another 30 days.
Is there a correlation between the amount of CDK4 / hTERT expression with the doubling of the population rate?
Table 3: what is the difference between “population doubling” and “doubling time”?
For the experiments carried out to test the efficiency of the lentiviral promoters, how can the authors assess that the plasmid is really expressed? They added the same resistant drug as the one used for immortalization. In this case, the immortalized cells are already resistant to puromycin, and it will not be possible to select the cells newly expressing the mCherry plasmid. How did the authors assess the efficiency of transduction for each plasmid?
Have the authors transduced DMD iHMuSCs? Will the effiency be the same?
Minor concerns:
Line 142, 165, 194: “iHMuSCs” instead “IHMuSCs”
Line 195: write “6 h” or “six hours”
Author Response
For qPCR experiments, how was the normalized gene chosen? How were the MIQE guidelines followed?
We added in the method section a series of details in accordance with the MIQE guidelines and added a reference for the house keeping gene, lines 211-21.
hTert and CDK4 were analysed by qRT-PCR. How did the author made the difference between a transcript and a potential plasmid DNA contamination that is still present in the infectious supernatant even several days after transduction?
The qRT-PCR was done several weeks after the infection with the lentiviruses, so after expansion of the cell population, first as a bulk population, then, as clonal cells. It is therefore unlikely that plasmid DNA are translated in those middle term cultures. We added some clarification about the transduction process in the method section (lines 125-7).
The authors mention the use of 2 lentiviruses for immortalization but only mention 1 resistant drug (Puromycin) to select their population. How can they be sure that the population is pure and immortalized by selecting a single gene?
At the time of infection, the population is not pure, but is a mix of cells that have inserted either no genes, or only CDK4 gene or only hTERTgene or both. It was reported in the literature that both genes are required for expansion of myogenic cells (ref 13, 14, 15, 16 in the manuscript). So, after puromycin selection, only hTERT transduced cells are present. cells lacking CDK4 gene were shown to not expand. Therefore, expansion of the cells is a mraker of the CDK4 integration in their genome. Two steps of expansion were made: the first after transduction, to amplify the "bulk" immortalized cell populations, then the second amplification from the single cell cultures. Precisions were added lines 241-3.
Could the authors detailed the plasmids and the detailed protocol used for immortalization knowing that it is not described in the citation they mentioned ?
We apology for this and have now added details about the plasmids and the mode of production of hTERT and CDK4 lentiviruses, lines 118-127.
What are the characteristics of the hMuSCs native population in terms of doubling the population in the short and long term?
Since the primary hMuSCs do not expand for a very long time (about 3-4 passages only), this is not possible to answer the reviewer's question. The primary cells were used as early as possible after their isolation from the muscle to perform the transduction/immortalization procedure.
In figure 1, C and D, how is the doubling time generated? Is it an average of the total doubling rate calculated over the life span? If yes, how long was the life span for each clones? Because, as shown in figure 1A, some clones were stopped at 14 days and another 30 days.
We apology for the confusion. The doubling time was calculated over the culture span. Stopping the cultures after the selection period after infection ranged from 14 to 30-40 days. The heterogeneity was due to different practices with time. The first clones we analyzed were stopped at late time points. Then, since we observed that the expanding cells did grow very regularly from the seeding time, we shortened that period. We observed that the cells either grew very well and regularly (and were therefore selected), or they did not expand at all (and were not further studied). Clarifications have been added in the method section lines 138-9 and in the resuts section, lines 241-3.
There is no possibility to calculate the life span of each clone since the immortalized cell life span is theorically infinite. The results presented in section 3.5 on long term cultures provide promising data about the capability of these cells to retain long-term myogenic properties with time.
Is there a correlation between the amount of CDK4 / hTERT expression with the doubling of the population rate?
We performed the analysis requested by the reviewer and the results are presented here below (we attach a file since the graph cannot be pasted here). There is no correlation between the expression level of the transgenes and the population doubling time of the cells.
Table 3: what is the difference between “population doubling” and “doubling time”?
The population doubling means the number of doublings the cells have gone through at the time of the analysis (CD56 expression and differentiation capacities). We added some clarifications in the legend of Table 3 to avoid any confusion.
For the experiments carried out to test the efficiency of the lentiviral promoters, how can the authors assess that the plasmid is really expressed? They added the same resistant drug as the one used for immortalization. In this case, the immortalized cells are already resistant to puromycin, and it will not be possible to select the cells newly expressing the mCherry plasmid. How did the authors assess the efficiency of transduction for each plasmid?
Have the authors transduced DMD iHMuSCs? Will the effiency be the same?
The reviewer is wright, we could not have used puromycin for selecting infected cells because they are already resistant. After titration of the viruses in HEK293T, we assumed that the efficiency of infection would be similar to that previously obtained in our lab for C2C12 myoblast infection, since all viruses have a similar backbone. Changes were made in Method section, lines 204-5.
We have not tried to transduce DMD iHMuSCs, though we have shown in our lab that primary DMD MuSCs are transduced as efficiently as WT MuSCs with the EF1a promoter lentivirus. Also the objective of this experiment was to test the promoter for a speficic cell type, i.e. IHMuSCs, assuming that DMD, WT and DMC derived cells are of the same nature. This was indicated line 373.
Minor concerns:
Line 142, 165, 194: “iHMuSCs” instead “IHMuSCs”
This was modified.
Line 195: write “6 h” or “six hours”
This was modified.

Round 2
Reviewer 1 Report
Authors answered to the reviewer's comments. There is no further comments or concerns on the revised Ms.
Reviewer 2 Report
The authors have addressed most of my concerns.
I am still not entirely convinced by the part concerning the activity of the promoter and the choice of the best in these experimental conditions, but the main message of this article is the previous characterization part, which is well designed.
congratulations.